# Real-Time and Ultrasensitive Prostate-Specific Antigen Sensing Using Love-Mode Surface Acoustic Wave Immunosensor Based on MoS_2_@Cu_2_O-Au Nanocomposites

**DOI:** 10.3390/s24237636

**Published:** 2024-11-29

**Authors:** Yan Yu, Haiyu Xie, Tao Zhou, Haonan Zhang, Chenze Lu, Ran Tao, Zhaozhao Tang, Jingting Luo

**Affiliations:** 1Shenzhen Key Laboratory of Advanced Thin Films and Applications, GuangDong Engineering Technology Research Centre of Breath Test, College of Physics and Optoelectronic Engineering, Shenzhen University, Shenzhen 518060, China; 2200451022@email.szu.edu.cn (Y.Y.); 1810342119@email.szu.edu.cn (H.X.); 2200453042@email.szu.edu.cn (T.Z.); 2300451008@email.szu.edu.cn (H.Z.); luojt@szu.edu.cn (J.L.); 2Key Laboratory of Specialty Agri-Products Quality and Hazard Controlling Technology of Zhejiang Province, College of Life Sciences, China Jiliang University, Hangzhou 310018, China; chenzelu@cjlu.edu.cn; 3Water Science and Environmental Engineering Research Centre, College of Chemistry and Environmental Engineering, Shenzhen University, Shenzhen 518060, China

**Keywords:** Love-mode surface acoustic wave (L-SAW), immunosensor, nanocomposite, prostate-specific antigen (PSA), real time, ultrasensitive

## Abstract

Prostate-specific antigen (PSA) is a well-established tumour marker for prostatic carcinoma. In this study, we present a novel, real-time, and ultrasensitive Love-mode surface acoustic wave (L-SAW) immunosensor for PSA detection enhanced by MoS_2_@Cu_2_O-Au nanocomposite conjugation. The MoS_2_@Cu_2_O-Au nanocomposites were analyzed by SEM, XRD, and EDS. The experiments show a significant improvement in sensitivity and detection limit compared with the previous detection methods utilizing nanogold alone to detect PSA biomolecules. The experimental results show a good linear relationship when the range of PSA concentrations between 200 pg/mL and 5 ng/mL was tested. The experimental results also show good specificity against alpha 1 fetoprotein and L-tryptophan disruptors.

## 1. Introduction

Despite significant advancements in biomedical sciences and technologies, cancer remains a leading cause of mortality worldwide [1]. Prostate cancer, in particular, exhibits one of the highest incidence rates among malignancies affecting males [2]. Its often asymptomatic nature poses significant challenges for timely diagnosis and treatment, contributing to its high mortality rate. Therefore, the early diagnosis of prostate cancer, particularly through the sensitive and precise detection of tumour markers [3], is crucial for improving patient outcomes. Prostate-specific antigen (PSA) is a well-established biomarker for prostate cancer screening and diagnosis [4,5]. Serum PSA concentrations exceeding 4 ng/mL are associated with a significantly elevated risk of prostate cancer [6]. Current immunoassay methods for PSA detection, such as fluorescence immunoassay [7], electrochemiluminescence [8], surface plasmon resonance [9], and electrochemical immunoassay [10], often suffer from limitations such as high cost, time-consuming procedures, cumbersome equipment, and a high risk of damaging the recognition elements. Therefore, there is a pressing need for the development of cost-effective, accurate, and portable PSA detection devices.

Meanwhile, surface acoustic wave (SAW) sensors have been used for immunoassays due to their mass-loading effect. Compared to other biosensing technologies mentioned above, SAW-based platforms are particularly well suited for detecting bio-trace analytes in liquid environments due to their real-time detection capabilities, high sensitivity, strong anti-interference properties, and low detection limits [11,12]. These advantages have spurred significant research interest in SAW-based biosensors for diverse applications. For instance, Gagliardi et al. developed a SAW-based lab-on-a-chip for the rapid detection of Legionella pneumophila in water [13], and Bharati et al. designed a DNA biosensor using an inverted Lamb wave SAW resonator based on ZnO/SiO_2_/Si/ZnO membranes [14]. Among the different modes of SAWs, Love-mode SAW (L-SAW) holds particular promise for PSA detection. L-SAW sensors are able to operate effectively in liquid environments, coupled with high sensitivity, potential for miniaturization, and real-time detection capabilities [11], which are ideal for developing advanced PSA immunosensors.

The effective utilization of L-SAW immunosensors for PSA detection necessitates a signal enhancement method to achieve high sensitivity. Increasing mass loading by binding nanoparticles to an antibody can significantly enhance the sensitivity of the L-SAW immunosensors, and various materials, such as magnetic nanoparticles, metal nanoparticles, and metal oxides, have been used to label antibodies [15,16,17]. Among these, gold nanoparticles (AuNPs) are widely employed in biosensing due to their excellent biocompatibility [18,19]. However, AuNPs suffer from poor stability and a tendency to aggregate in electrolyte solutions [20,21]. To address these issues, researchers have developed metal oxide nanocomposites, such as Cu_2_O, by combining two or more nanomaterials through covalent bonds or electrostatic adsorption [11,17,22,23]. Gold–metal oxide nanocomposites used in biosensing not only retain the properties of the individual nanomaterials but also exhibit enhanced performance due to synergistic effects during preparation. Therefore, employing gold–metal oxide nanocomposites can further improve the performance of biosensors.

Among various nanomaterials, transition metal chalcogenides have emerged as promising candidates for biosensor applications due to their unique crystal and electronic structures [24]. Among them, molybdenum disulphide (MoS_2_), a layered nano-lamellar material with a large specific surface area and abundant active sites, can support the formation of diverse nanocomposites, further enhancing sensor performance [25]. For instance, Jing et al. [26] demonstrated highly sensitive thrombin detection using an electrochemical sensor based on poly (dimethyl diallyl ammonium chloride) (PDDA)/graphene/MoS_2_ nanocomposites. Similarly, Su et al. [27] achieved the sensitive detection of thrombin and adenosine triphosphate (ATP) using an electrochemical sensor enhanced by MoS_2_-Au nanocomposites. These studies highlight the potential of MoS_2_-based nanocomposites for developing high-performance L-SAW PSA immunosensors.

In this study, we report a real-time and ultrasensitive L-SAW immunosensor for PSA detection. The L-SAW immunosensor chip was designed using finite element analysis (FEA). Molybdenum disulphide@cuprous oxide (MoS_2_@Cu_2_O)–Au nanocomposites, synthesized by the hydrothermal method, were applied to enhance the mass loading effect. Sensor performance, including sensitivity, limit of detection, and specificity, was evaluated by monitoring changes in insertion loss, demonstrating significant potential for clinical applications in real-time PSA detection and prostate cancer diagnosis.

## 2. Materials and Methods

### 2.1. Materials and Instruments

Ammonium tetrathiomolybdate ((NH_4_)_2_MoS_4_) and copper nitrate trihydrate (Cu(NO_3_)_2_·3H_2_O) were purchased from Macklin Biochemical Technology Co., Ltd. (Shanghai, China). Hydrazinehydrate (N_2_H_4_·H_2_O) and 3-Mercaptopropionicacid (MPA) were purchased from Aladdin Biochemical Technology Co., Ltd. (Shanghai, China). Tetrachloroauric acid trihydrate (HAuCl_4_) was purchased from Toyong Bio Tech. Inc. (Shanghai, China). Sodium citrate (Na_3_C_6_H_5_O_7_) was purchased from Yuanye Bio-Technology Co., Ltd. (Shanghai, China). N-Hydroxy succinimide (NHS), phosphate-buffer saline (PBS) solution, and 1-(3-Dimethylaminopropyl)-3-ethylcarbodiimide (EDC) were purchased from Yuduo Bio-Technology Co., Ltd. (Shanghai, China). PSA, alpha-fetoprotein (AFP), carcinoembryonic antigen (CEA), L-tryptophan, PSA primary antibodies (Ab_1_), and PSA secondary antibodies (Ab_2_) were purchased from Biorbyt (Cambridge, UK). Bovine Serum Albumine (BSA) was purchased from Sigma-Aldrich (St. Louis, MI, USA).

The surface morphologies of AuNPs, Cu_2_O, MoS_2_ and MoS_2_@Cu_2_O-Au nanocomposites were characterized using a field emission scanning electron microscope (FE-SEM) (Supra 55 Sapphire, Zeiss, Oberkochen, Germany). A high-resolution transmission electron microscope (HRTEM) (Talos F200i, Thermo Fisher, Waltham, MA, USA) and X-ray diffraction (XRD) (X’pert pro, PANalytical, Almelo, The Netherlands) with Cu Kα radiation source were used to investigate the crystalline structures of the nanocomposites.

### 2.2. Numerical Modelling of L-SAW Immunosensor

A simplified two-dimensional (2D) FEA model of the L-SAW immunosensor chip was developed using COMSOL Multiphysics 6.0. The model was constructed by utilizing the modules “Solid Mechanics” and “Electrostatics”, coupled through the module “Piezoelectric Effect”. And the global displacement u and electric potential V were derived by solving Equations (A6) and (A7) in Appendix A using a frequency-based solver. The 2D computational domain of the model, as depicted in Appendix A, comprised a 36° YX-cut LiTaO_3_ substrate (17050 μm × 125 μm), coated with a 1 μm SiO_2_ waveguide layer. A 9 mm wide sensitive area was positioned atop the SiO_2_ waveguide layer. A 25 μm thick perfectly matched layer (PML) was applied to the bottom, left, and right boundaries of the model, while all boundaries were designated as free. Input and output interdigital transducers (IDTs) consisted of 40 finger pairs each, with an electrode width and pitch of 6.25 μm. Since the thickness of the Au sensitive area and electrodes (100 nm) was negligible compared to the waveguide layer, their mechanical influence was excluded from the model. A sinusoidal signal of 1 V was applied to the input IDT, while the output IDT was in the forest to floating potential condition. A mapping method with a maximum mesh size of λ/8 (3.125 μm) was employed to ensure simulation accuracy while minimizing computational complexity.

### 2.3. Preparation of L-SAW Sensor Chip

The fabricated L-SAW sensor chip is shown in Figure 1a, with detailed parameters provided in Appendix A. A pair of IDTs was patterned on a 36° YX-cut LiTaO_3_ substrate with a delay line structure using conventional photolithography and lift-off processes. A 1 μm thick SiO_2_ waveguide layer was deposited via magnetron sputtering over the IDTs and delay line region. Finally, a 100 nm Au film was deposited via thermal evaporation onto the SiO_2_ waveguide layer within the designated sensitive area.

### 2.4. Preparation of MoS_2_@Cu_2_O-Au-Ab_2_ Conjugates

MoS_2_@Cu_2_O nanocomposites were synthesized via a facile, one-step hydrothermal method [28], as illustrated in Figure 1b. (NH_4_)_2_MoS_4_ and Cu(NO_3_)_2_·3H_2_O were reacted in a Teflon-lined stainless-steel autoclave at 200 °C for 10 h in the presence of the reducing agent N_2_H_4_·H_2_O, yielding MoS_2_@Cu_2_O nanocomposites. For comparison, Cu_2_O nanomaterials and MoS_2_ nanoflowers were synthesized using a procedure similar to that employed for the MoS_2_@Cu_2_O nanocomposites. However, the synthesis of Cu_2_O nanomaterials was conducted without the inclusion of (NH_4_)_2_MoS_4_, and, similarly, the synthesis of MoS_2_ nanoflowers did not incorporate Cu(NO_3_)_2_·3H_2_O. The synthesis of AuNPs followed a published method [29], wherein a sodium citrate solution was added to a boiling HAuCl_4_ solution and heated for 20 min. After cooling to room temperature, a deep red solution of AuNPs was obtained. Subsequently, the solution containing MoS_2_@Cu_2_O nanocomposites was mixed and stirred for 24 h with the AuNP solution to prepare MoS_2_@Cu_2_O-Au nanocomposites. Finally, the MoS_2_@Cu_2_O-Au dispersion was mixed with a solution of Ab_2_ and stirred at 4 °C for 10 h. Following centrifugation, the MoS_2_@Cu_2_O-Au-Ab_2_ conjugates were obtained.

### 2.5. Experimental Framework Setup

The experimental setup, displayed in Figure 1c, consists of two main components: sample injection and data acquisition. A multi-channel programmable microfluidic pumping system was employed for sample injection. Solutions of PBS, PSA, and MoS_2_@Cu_2_O-Au-Ab_2_ mixture were loaded into separate precisely controlled syringes. A three-port to one-port inflow joint directed the solutions into one microchannel aligned with the sensitive area of the L-SAW immunosensor chip. The data acquisition system involved placing the L-SAW immunosensor chip within a reaction chamber maintained at room temperature (25 °C). The input and output IDTs of the chip were connected to a vector network analyzer (VNA), interfaced with a real-time data acquisition programme for signal recording. All sample solutions were injected at a controlled flow rate of 1 mL/h. Initially, PBS was first injected onto the surface of the sensitive area within the chamber. After approximately 10 min, during which, the frequency response of the device stabilized, a pre-prepared solution of PSA and MoS_2_@Cu_2_O-Au-Ab_2_ mixture was injected onto the surface of the sensitive area. The sample injection duration was approximately 25 min to ensure sufficient interaction time between the analyte and the immobilized biomolecular element, while also preventing any potential leakage. After Ab_1_ and PSA no longer bound to each other, the frequency began to stabilize, and the test concluded after a designated duration, with the entire test lasting approximately 50 min.

### 2.6. Construction of the L-SAW Immunosensor

The L-SAW immunosensor functionalization process is illustrated in Figure 2. First, the sensor surface was treated with an MPA solution. MPA molecules self-assembled onto the gold surface of the sensitive area via gold–thiol (Au-S) bonding, providing a stable platform for Ab_1_ immobilization. Next, a PBS solution containing the Ab_1_ specific to PSA was introduced, allowing Ab_1_ immobilization onto the MPA layer. Subsequently, BSA was employed to block any remaining active sites on the MPA layer, minimizing non-specific binding. Finally, the target solution containing PSA was prepared by complexing PSA with Ab_2_ conjugated to MoS_2_@Cu_2_O-Au nanocomposites. Subsequently, PSA in the target solution was captured by the Ab_1_ immobilized on the sensitive area surface. This functionalization strategy created a sandwich-like sensing structure (Figure 2) on the L-SAW immunosensor chip.

## 3. Results and Discussion

### 3.1. FEA Simulations and Electroacoustic Response of L-SAW Devices

Initially, the impact of incorporating a SiO_2_ waveguide layer on the vibration modes and insertion loss characteristics of the sensor was examined through FEA simulation. Appendix A illustrates that, for a constant input voltage, the incorporation of the SiO_2_ waveguide layer not only enhances the kinetic energy density (with the maximum value increasing from 11.8 J/m^3^ to 12.3 J/m^3^), but also further concentrates the vibrations at the surface of the device (where the depth of propagation is reduced from twice the wavelength to the wavelength itself). This modification is expected to improve the sensor’s sensitivity to variations in surface loading, thereby contributing to an increase in both the sensitivity and detection limit of the sensor. This phenomenon can be attributed to the waveguide effect of SiO_2_, which is characterized by a lower speed of sound compared to LiTaO_3_ [30]. This effect results in a transition of the vibration mode from a leaky SAW to a Love-mode SAW, thereby leading to a greater concentration of wave energy at the surface of the device. Appendix A compares the impact of incorporating the SiO_2_ waveguide layer on the theoretical insertion loss characteristics of the sensor. The results derived from FEA simulations indicate that the addition of the SiO_2_ waveguide layer alters the insertion loss from −14.22 dB to −11.61 dB, representing an improvement of approximately 2.6 dB, which effectively enhances the received signal. Furthermore, the insertion loss curves obtained from subsequent experimental measurements, as depicted in Appendix A, corroborate the findings from the FEA simulations, with the insertion loss varying from −13.64 dB to −11.38 dB (approximately 2.3 dB). This further substantiates the assertion that the introduction of the SiO_2_ waveguide layer significantly enhances the sensitivity of the sensor.

Subsequently, FEA simulation was employed to investigate the vibration patterns and to predict the insertion loss characteristics the L-SAW immunosensor. The simulated amplitude spectrum of the *Y*-direction displacement on the sensor surface (Figure 3a) exhibits multiple peaks within the 150–200 MHz range, with the dominant peak located near 173 MHz. This indicates that the primary displacement of surface particles occurs along the *Y*-axis, consistent with the expected behaviour of a Love wave. The displacement field distribution at centre frequency (Figure 3b) further confirms the Love wave propagation mode, characterized by predominantly transverse particle motion concentrated near the device surface. This confinement of acoustic energy to the surface is advantageous for achieving high sensitivity in liquid environments. Figure 3c compares the theoretical and experimentally measured insertion loss curves of the L-SAW immunosensor. The FEA simulation predicts a centre frequency of 174.165 MHz and an insertion loss of −11.89 dB. Experimental measurements yielded a centre frequency of 173.405 MHz and an insertion loss of −11.61 dB. The minor discrepancies between the simulated and experimental results are attributed to factors such as variations in the deposited film thicknesses during fabrication and the simplification of the FEA model, which omitted the thin Au film. To assess the response of L-SAW to liquid loading, the insertion loss was monitored before and after the introduction of PBS solution (Figure 3d). A shift in insertion loss from −18.62 dB to −19.46 dB (approximately 0.8 dB) was observed upon PBS injection, attributable to the increased pressure exerted by the microfluidic channel on the sensitive area of the L-SAW immunosensor. These findings demonstrate the feasibility of employing the Au-coated L-SAW immunosensor for biomolecule detection in liquid environments.

### 3.2. Characterization of MoS_2_@Cu_2_O-Au Nanocomposites

The morphology and microstructure of the synthesized nanomaterials were characterized using SEM (Figure 4). As shown in Figure 4a, the gold nanoparticles exhibit a spherical or quasi-spherical morphology with a uniform size distribution around 30 nm. The nanoparticles are well dispersed, indicating minimal aggregation. Figure 4b reveals the characteristic sheet-like structure of MoS_2_, forming a three-dimensional porous network composed of randomly stacked two-dimensional nanosheets. This unique morphology provides a high surface area and abundant binding sites. Figure 4c presents the structure of Cu_2_O nanomaterials, which display a wider size distribution ranging from 100 to 200 nm and a variety of morphologies, including spherical, polyhedral, and nanorod structures. Similarly to MoS_2_, the Cu_2_O nanomaterials possess a large specific surface area and numerous binding sites. Figure 4d displays the MoS_2_@Cu_2_O-Au nanocomposites, which exhibit a distinctive coral-like morphology. This three-dimensional spatial coral structure, formed by the interconnected MoS_2_@Cu_2_O framework decorated with well-dispersed gold nanoparticles, further increases the specific surface area and provides abundant binding sites. The porous nature of the nanocomposites facilitates the efficient capture and binding of target PSA biomolecules, highlighting their suitability for biosensing applications.

The crystalline structure and phase purity of the synthesized nanomaterials were investigated using XRD. As depicted in Figure 5a, the XRD pattern of the MoS_2_ nanosheets (blue curve) exhibits characteristic diffraction peaks at 14.37°, 39.53°, 49.78°, and 58.3°, which correspond to the (002), (100), (103), and (110) planes, respectively, of hexagonal MoS_2_ (JCPDS No. 37-1492). The XRD pattern of the Cu_2_O nanomaterials (pink curve) displays prominent peaks at 42.6° and 74.4°, indexed to the (200) and (311) planes of cubic Cu_2_O (JCPDS No. 34-1354). The XRD pattern of the MoS_2_@Cu_2_O nanocomposites (green curve) exhibits diffraction peaks corresponding to both MoS_2_ and Cu_2_O, confirming the successful formation of the composite material. High-resolution transmission electron microscopy (HR-TEM) provided further insights into the microstructure of the MoS_2_@Cu_2_O-Au nanocomposites. Figure 5b and 5c reveal distinct lattice fringes with spacings of 0.64 nm, 0.44 nm, and 0.22 nm, corresponding to the (002) plane of MoS_2_, the (200) plane of Cu_2_O, and the (111) plane of gold nanoparticles, respectively. These observations confirm the intimate contact and adhesion of the gold nanoparticles onto the MoS_2_@Cu_2_O surface. Additionally, energy-dispersive X-ray spectroscopy (EDS) analysis (Appendix A) confirmed the presence of Mo, S, O, Cu, and Au signals, providing further evidence for the successful synthesis of the MoS_2_@Cu_2_O nanocomposites.

### 3.3. Sensing Performances of the L-SAW Immunosensor

The real-time PSA detection capability of the MoS_2_@Cu_2_O-Au-enhanced L-SAW immunosensor was evaluated using a series of PSA solutions with varying concentrations (Figure 6a). Initially, PBS buffer solution was flowed over the sensing area at a rate of 1 mL/h to establish a stable baseline signal. After 10 min, the injection solution was switched to a mixture of PSA and MoS_2_@Cu_2_O-Au nanocomposites, maintaining the same flow rate. This allowed the target PSA molecules to interact with the immobilized antibodies on the sensor surface for 25 min, ensuring sufficient time for binding. As shown in Figure 6a, the binding of the PSA–nanocomposite complex to the sensor surface resulted in a concentration-dependent decrease in the resonant frequency of the L-SAW device. The frequency shift increased with increasing PSA concentration, demonstrating a dynamic range of 0.2-50 ng/mL. At a PSA concentration of 50 ng/mL, a significant frequency shift of approximately −8000 Hz was observed, while a measurable response of −500 Hz was detected at the lowest tested concentration of 0.2 ng/mL. These results highlight the enhanced sensitivity and lower detection limit achieved by incorporating MoS_2_@Cu_2_O-Au nanocomposites compared to the control group employing AuNPs alone for PSA detection (Figure 6b). The significant amplification in signal response can be attributed to the larger mass loading and increased surface area provided by the MoS_2_@Cu_2_O-Au nanocomposite labels.

To further investigate the relationship between PSA concentration and sensor response, FEA simulations were performed to predict the theoretical frequency shift in the L-SAW immunosensor at various PSA concentrations, assuming a 1:20 ratio of nanogold to PSA (Appendix A). As anticipated, the simulations revealed a linear correlation between frequency shift and PSA concentration, reflecting the proportional relationship between mass loading and surface stress. Experimental validation of the sensing performance was conducted using MoS_2_@Cu_2_O-Au nanocomposites. Figure 6c displays the measured frequency shifts for different PSA concentrations. Consistent with the simulations, a linear response was observed within the range of 200 pg/mL to 5 ng/mL. The linear fitting within this range yielded a slope of −1026.8 Hz/(ng/mL) and a correlation coefficient (R^2^) of 0.9609, closely matching the simulated slope of −1029.2 Hz/(ng/mL) (R^2^ = 0.9999). This agreement supports the validity of the 1:20 nanogold-to-PSA ratio used in the simulations. However, at PSA concentrations exceeding 10 ng/mL, the experimentally observed frequency shift plateaued, deviating from the linear trend predicted by the simulations (Appendix A and Figure 6c). This discrepancy is attributed to the limited binding capacity of the immobilized antibodies on the sensor surface. The simulation assumed complete capture of all PSA-nanocomposite complexes, while in reality, the number of available antibody binding sites is finite.

Based on the experimental data, the limit of detection (LOD) for PSA was determined to be 76 pg/mL, calculated as three times the root mean square deviation (RMSD) of the background noise. This high sensitivity highlights the potential of this L-SAW immunosensor for detecting low concentrations of PSA. The analytical performance of the L-SAW immunosensor, such as linear range and detection limit, is comparable to or superior to that of other immunosensors employed for PSA detection, as demonstrated in Table 1. Although Raman frequency shift-based immunosensors, thin-film body acoustic resonator-based immunosensors, and electrochemical immunosensors exhibit broader linear ranges than our method, our proposed L-SAW immunosensor has a better detection limit compared to the other assays and facilitates rapid, real-time analysis.

The specificity of the L-SAW PSA immunosensor was evaluated by challenging the sensor with potential interfering agents, L-tryptophan, AFP, and CEA, at a repeated concentration of 50 ng/mL 50 times. As shown in Figure 6d, the frequency shifts induced by these disruptors (770 ± 25 Hz for L-tryptophan, 820 ± 35 Hz for AFP, and 810 ± 30 Hz for CEA) were significantly lower than the response elicited by 50 ng/mL PSA (7980 ± 85 Hz). This demonstrates the high specificity of the immobilized antibodies for PSA, minimizing the potential for false-positive signals.

## 4. Conclusions

In summary, we have presented the design, fabrication, and characterization of a novel L-SAW immunosensor based on a 36° YX-cut LiTaO_3_ wafer for the real-time detection of PSA with high sensitivity and specificity. The sensor utilizes MoS_2_@Cu_2_O-Au nanocomposites as signal amplification labels to enhance the detection of low PSA concentrations. Within the low concentration range of 0.2 ng/mL to 5 ng/mL, the immunosensor demonstrates favourable linear performance in PSA detection, aligning with the frequency shift trends calculated by the simulation results. In comparison to immunosensors that utilize only AuNPs, the L-SAW immunosensor employing MoS_2_@Cu_2_O-Au nanocomposites exhibits superior sensitivity of 1026.8 Hz/(ng/mL), with an LOD of 76 pg/mL. Additionally, this immunosensor shows excellent selectivity for PSA, exhibiting minimal affinity for common tumour marker proteins such as L-tryptophan, AFP, and CEA. Therefore, the proposed L-SAW immunosensor is expected to serve as an ideal tool for medical applications, particularly in the early-stage screening and diagnosis of prostate cancer.

## Figures and Tables

**Figure 1 sensors-24-07636-f001:**
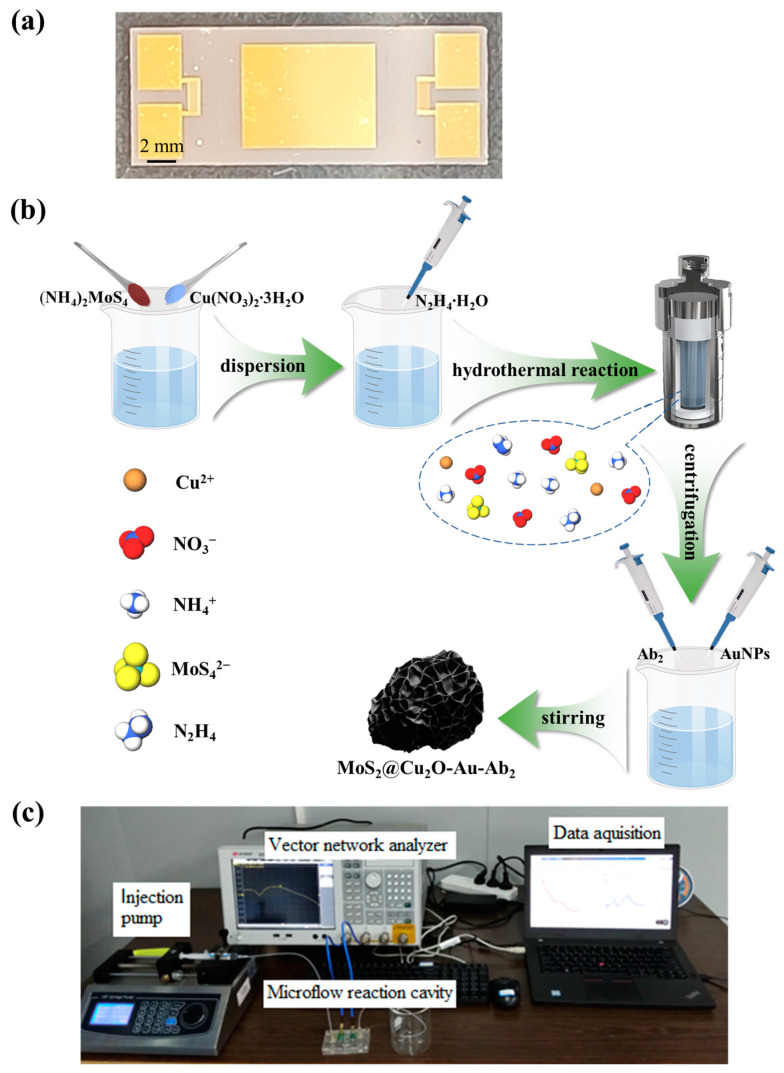
(**a**) Photo of the fabricated L-SAW sensor chip; (**b**) synthesis process of MoS_2_@Cu_2_O-Au-Ab_2_ conjugates; (**c**) photo of the experimental setup.

**Figure 2 sensors-24-07636-f002:**
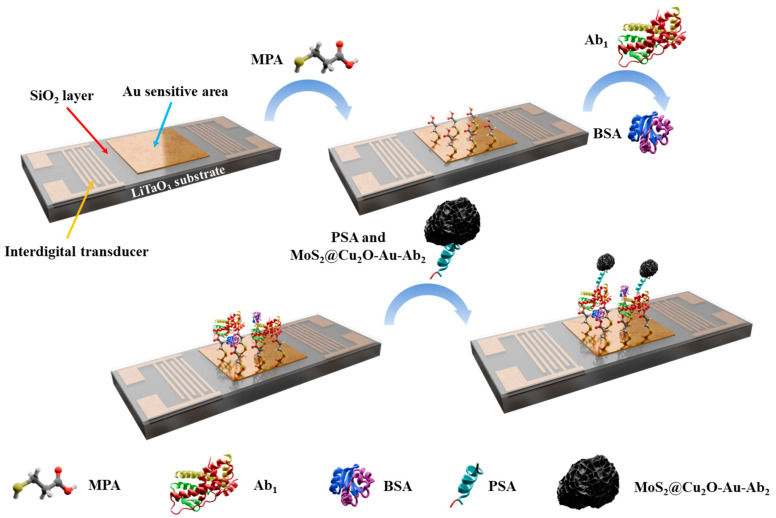
Biosensing steps of L-SAW PSA immunosensor utilizing MoS_2_@Cu_2_O-Au nanocomposites.

**Figure 3 sensors-24-07636-f003:**
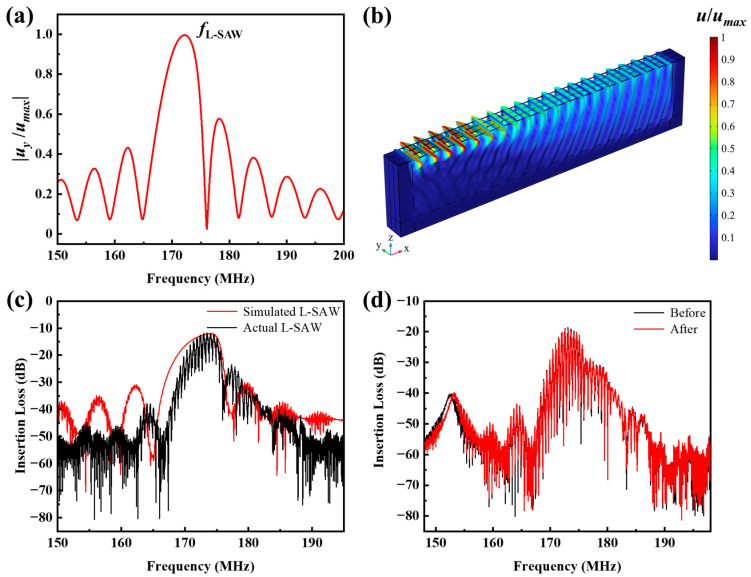
(**a**) *Y*-direction displacement amplitude spectrum of the L-SAW immunosensor surface obtained by FEA simulation; (**b**) normalized displacement field of the L-SAW immunosensor under peak frequency excitation; (**c**) FEA-simulated S_21_ signals and actual S_21_ signals of the L-SAW immunosensor; (**d**) S_21_ signals of the fabricated L-SAW immunosensor before and after injection of PBS solution.

**Figure 4 sensors-24-07636-f004:**
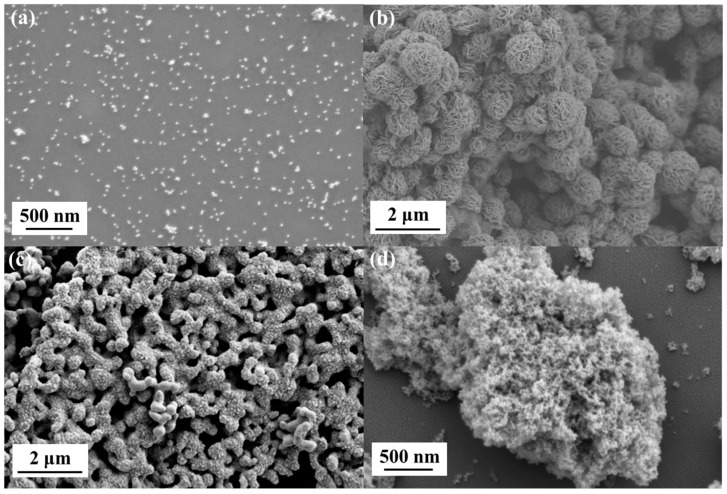
SEM images of (**a**) AuNPs; (**b**) MoS_2_; (**c**) Cu_2_O; (**d**) MoS_2_@Cu_2_O-Au.

**Figure 5 sensors-24-07636-f005:**
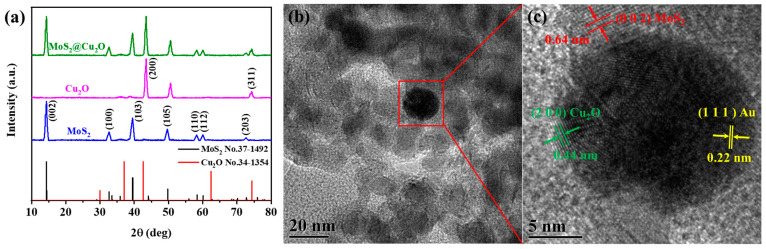
(**a**) XRD patterns of MoS_2_, Cu_2_O, and MoS_2_@Cu_2_O; (**b**) and (**c**) HR-TEM analysis of MoS_2_@Cu_2_O-Au.

**Figure 6 sensors-24-07636-f006:**
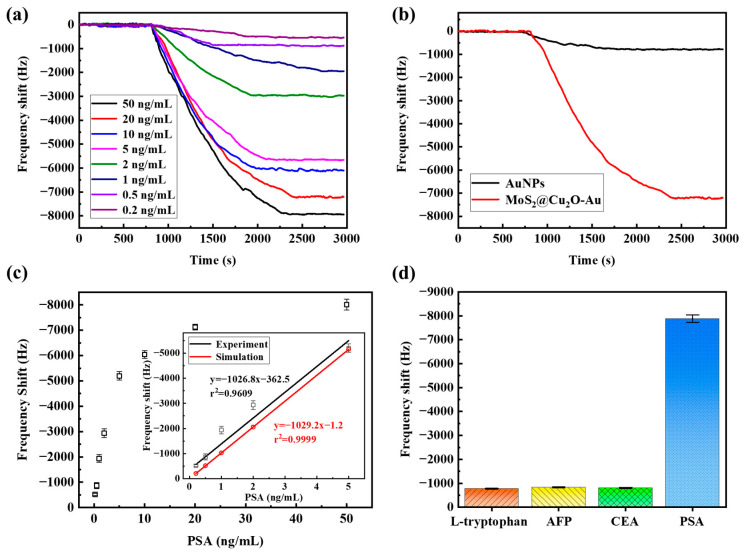
(**a**) Real-time responses of the L-SAW immunosensor to PSA with the concentration varying from 0.2 ng/mL to 50 ng/mL; (**b**) comparison of real-time responses of L-SAW immunosensors using AuNPs alone and MoS_2_@Cu_2_O-Au nanocomposites at a PSA concentration of 20 ng/mL; (**c**) frequency shifts corresponding to the various PSA concentrations, ranging from 0.2 ng/mL to 50 ng/mL; experimental results (black line) and simulation results (red line) demonstrate a linear relationship between the frequency shifts and the PSA concentration, spanning from 0.2 ng/mL to 5 ng/mL; (**d**) the selectivity analysis of the L-SAW immunosensor to 50 ng/mL solutions of L-tryptophan, AFP, CEA, and PSA.

**Table 1 sensors-24-07636-t001:** Comparison of other immunosensors for detection of PSA with L-SAW immunosensor.

Technique	Linear Range	Detection Limit	References
A novel split mode TFBAR device for quantitative measurements of prostate specific antigen in a small sample of whole blood	0–10 ng/mL	340 pg/mL	[31]
An electrochemical biosensor for prostate cancer biomarker detection using graphene oxide-gold nanostructures	0.2–10 ng/mL	200 pg/mL	[32]
SERS-based biosensor for detection of f-PSA%: Implications for the diagnosis of prostate cancer	1.0–200 ng/mL	0.7 ng/mL	[33]
Comparative study of aptasensor vs. immunosensor for label-free PSA cancer detection on GQDs-AuNRs modified screen-printed electrodes	0–11.6 ng/mL	140 pg/mL	[34]
3D label-free prostate specific antigen (PSA) immunosensor based on graphene-gold composites	0–10 ng/mL	0.59 ng/mL	[35]
Biocompatible osmium telluride-polypyrrole nanocomposite material: application in prostate specific antigen immunosensing	0–15 ng/mL	360 pg/mL	[36]
Detection of early stage prostate cancer by using a simple carbon nanotube@paper biosensor	0–500 ng/mL	1.18 ng/mL	[37]
Futuristic silicon photonic biosensor with nanomaterial enhancement for PSA detection	2.5–11 ng/mL	170 pg/mL	[38]
Real-time and ultrasensitive prostate-specific antigen sensing using Love-mode surface acoustic wave immunosensor based on MoS_2_@Cu_2_O-Au nanocomposites	0.2–5 ng/mL	76 pg/mL	This work

## Data Availability

The original contributions presented in the study are included in the article and Appendix A, further inquiries can be directed to the corresponding author.

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
