# Peer review of "Real-Time and Ultrasensitive Prostate-Specific Antigen Sensing Using Love-Mode Surface Acoustic Wave Immunosensor Based on MoS2@Cu2O-Au Nanocomposites"

_sensors, 2024, doi:10.3390/s24237636_

Round 1
Reviewer 1 Report
Comments and Suggestions for Authors
In this paper, authors proposes an L-SAW immunosensor for PSA detection using MoS2@Cu2O-Au nanocomposites composites as a signal amplification label. after careful review, I don't feel it's appropriate to publish in this journal:
1.The innovation of this paper is seriously insufficient. The sensor constructed in this study uses the signal tag material MoS2@Cu2O-Au nanocomposite, which is the same material used by the author's team in the paper published in 2021, but the detection substance of the sensor has changed, and the sensor performance has not improved significantly.(Rauf, S.; Qazi, H.I.A.; Luo, J.; Fu, C.; Tao, R.; Rauf, S.; Yang, L.; Li, H.; Fu, Y. Ultrasensitive Leaky Surface Acoustic Wave Immunosensor for Real-Time Detection of Alpha-Fetoprotein in Biological Fluids. Chemosensors 2021, 9, 311.)
Furthermore, the synthesis method of MoS2@Cu2O-Au nanocomposites in this paper is highly similar to that of a literature published in Sensors and Actuators B: Chemical in 2019.(Ning Ma, Tong Zhang, Dawei Fan, Xuan Kuang, Asghar Ali, Dan Wu, Qin Wei, Triple amplified ultrasensitive electrochemical immunosensor for alpha fetoprotein detection based on MoS2@Cu2O-Au nanocomposites, Sensors and Actuators B: Chemical, Volume 297, 2019, 126821.), and even the schematic diagram of the composite material is basically the same as that in the literature.
2.There are some other problems in the paper, such as: the schematic diagrams of MoS2@Cu2O-Au nanocomposites in Figure 1b and Figure 2 are inconsistent; the morphology image of MoS2@Cu2O-Au nanocomposites in Figure 4d is not clear enough.
3.The English of the paper also needs to be further improved.
Reviewer 2 Report
Comments and Suggestions for Authors
In this manuscript “Real-Time and Ultrasensitive Prostate-Specific Antigen Sens-2 ing Using Love-Mode Surface Acoustic Wave Immunosensor 3 Based on MoS2@Cu2O-Au Nanocomposites”, Yu and co-workers described a novel sensing platform that uses MOS2@Cu2O-Au nanocomposites for amplification signal in a L-SAW immunosensor for real time PSA detection. While the idea seemed novel and interesting, more data are required to support the claims :
- In Figure 2 and section 2.6., the authors explained the different steps involved in the biosensor detection. However, there is no references about the time and incubations conditions of the targe solution (line 163-164, complex od PSA with secondary antibody conjugated to MOS2@Cu2O-Au nanocomposites. Under what conditions does this conjugation take place? Is this conjugation performed off-line and then injected, or is it performed in the reaction chamber after each component has been injected separately?
- The authors also explained that all the sample solution were injected a controlled flow rate to ensure sufficient interaction time between the analyte and the surface of the immunosensor chip. How long does the analyte solution pass through the reaction chamber? I also suggest explaining how much time is needed to complete the whole process.
- In Section 3.2, the authors characterize the synthesized nanomaterials: AuNPs, MoS2, Cu2O and MOS2@Cu2O-Au nanocomposites. However, only the synthesis of AuNps and MOS2@Cu2O-Au nanocomposites is described in the previous sections. How are the other nanomaterials synthesized? 1. What is the purpose of these two nanomaterials (MoS2 and Cu2O) As far as I understand, these NMs have not been mentioned before and have no function within the immunosensor.
- In Figure 4, the authors says that Figure 4b corresponds to Cu2O and Figure 4c corresponds to MoS2. But, in section 3.2 lines 203-207, the nomenclature for Figures 4b and 4c is reversed. Which is the correct image that corresponds to each type of nanomaterial?
- The authors tested the specificity of the biosensor selecting two interfering agents: alpha-1 Fetoprotein and L-tryptophan. What are the reasons for selecting these potential interferences? Can there be other species that affect the sensor response?
- In relation to the previous comment, the authors claimed that the biosensor would be used for PSA quantification in clinical applications and prostate cancer detection. However, there is no real sample analysed to validate the methodology and demonstrate that there will be any effect on sensitivity due to the complex matrix present in a real serum sample. It would be convenient to test the innmunosensor in real samples.
- It is reported in the conclusions section that the use of MoS2@Cu2O-Au nanocomposites to enhance the detection of PSA is more sensitive than the sensitivity achieved by simplest AuNPs based biosensors. However, the sensitivity of these sensors is not specified. I think that the direct comparison of the two sensitivities could be valuable.
Round 2
Reviewer 1 Report
Comments and Suggestions for Authors
The article "Real-Time and Ultrasensitive Prostate-Specific Antigen Sensing Using Love-Mode Surface Acoustic Wave Immunosensor Based on MoS2@Cu2O-Au Nanocomposites "(sensors-3297660) can be acceptable.
Reviewer 2 Report
Comments and Suggestions for Authors
The authors have carefully addressed the comments and revised the manuscript increasing the number of references to strengthen the background and for comparison with other methods. In addition, new experimental results have been included by evaluating another possible interfering agent (CEA). I suggest the acceptance of this manuscript for publication in Sensors.
Author Response
We would like to express our sincere gratitude for the time you dedicated to reviewing our manuscript. Additionally, we greatly appreciate your acknowledgment of our research.